# Comparable in vivo joint kinematics between self-reported stable and unstable knees after TKA can be explained by muscular adaptation strategies: A retrospective observational study

**Longfeng Rao[1], Nils Horn[2], Nadja Meister[1], Stefan Preiss[2], William R Taylor[1]\*, Alessandro Santuz[1,3], Pascal Schütz[1]**

[1]Laboratory for Movement Biomechanics, Institute for Biomechanics, Zurich, Switzerland; [2]Department of lower extremities, Schulthess Clinic Zurich, Zurich, Switzerland; [3]Max Delbrück Center for Molecular Medicine, Berlin, Germany

**\*For correspondence:**
bt@ethz.ch

**Competing interest:** The authors declare that no competing interests exist.

## Abstract

**Background:** Postoperative knee instability is one of the major reasons accounting for unsatisfactory outcomes, as well as a major failure mechanism leading to total knee arthroplasty (TKA) revision. Nevertheless, subjective knee instability is not well defined clinically, plausibly because the relationships between instability and implant kinematics during functional activities of daily living remain unclear. Although muscles play a critical role in supporting the dynamic stability of the knee joint, the influence of joint instability on muscle synergy patterns is poorly understood. Therefore, this study aimed to understand the impact of self-reported joint instability on tibiofemoral kinematics and muscle synergy patterns after TKA during functional gait activities of daily living.

**Methods:** Tibiofemoral kinematics and muscle synergy patterns were examined during level walking, downhill walking, and stair descent in eight self-reported unstable knees after TKA (3M:5F, 68.9 ± 8.3 years, body mass index [BMI] 26.1 ± 3.2 kg/m², 31.9 ± 20.4 months postoperatively), and compared against 10 stable TKA knees (7M:3F, 62.6 ± 6.8 years, 33.9 ± 8.5 months postoperatively, BMI 29.4 ± 4.8 kg/m²). For each knee joint, clinical assessments of postoperative outcome were performed, while joint kinematics were evaluated using moving video-fluoroscopy, and muscle synergy patterns were recorded using electromyography.

**Results:** Our results reveal that average condylar A-P translations, rotations, as well as their ranges of motion were comparable between stable and unstable groups. However, the unstable group exhibited more heterogeneous muscle synergy patterns and prolonged activation of knee flexors compared to the stable group. In addition, subjects who reported instability events during measurement showed distinct, subject-specific tibiofemoral kinematic patterns in the early/mid-swing phase of gait.

**Conclusions:** Our findings suggest that accurate movement analysis is sensitive for detecting acute instability events, but might be less robust in identifying general joint instability. Conversely, muscle synergy patterns seem to be able to identify muscular adaptation associated with underlying chronic knee instability.

**Funding:** This research received no specific grant from any funding agency in the public, commercial, or not-for-profit sectors.

## Editor's evaluation

This paper presents a new method for evaluating joint instability at the knee after total knee replacement. The work is a valuable contribution that is based on solid evidence. The results should be of particular interest to those who study this important clinical problem.

## Introduction

Postoperative knee instability is one of the major reasons accounting for unsatisfactory outcomes, as well as a major failure mechanism leading to revision surgery after primary total knee arthroplasty (TKA). However, postoperative knee instability is not well defined clinically, and the boundaries between stable and unstable TKA knees are still poorly understood. This lack of understanding is likely due to the multitude of factors that are associated with knee instability, including inadequate soft-tissue balancing (*Hosseini Nasab et al., 2019*; *Song et al., 2014*), loss of ligamentous integrity (*Nagle and Glynn, 2020*), and improper component sizing (*Nagle and Glynn, 2020*; *Parratte and Pagnano, 2008*).

Clinical assessment of joint stability relies on passive laxity examinations and patient-reported outcome measures (PROMs). Manual stress tests, including the anterior/posterior drawer, Lachman, and varus/valgus tests, as well as questionnaires such as the Knee injury and Osteoarthritis Outcome Score (KOOS) (*Roos et al., 1998*), Oxford Knee Score (OKS) (*Dawson et al., 1998*), University of California and Los Angeles (UCLA) Activity Scale (*Zahiri et al., 1998*), and the Western Ontario and McMaster Universities Osteoarthritis Index (WOMAC) (*Bellamy et al., 1988*) are widely used in clinical practice to assess knee instability after TKA. However, such clinical assessments are highly subjective with large inter-rater variability (*Mears et al., 2022*). With the goal to objectively investigate knee instability, methods for quantifying passive knee laxity such as the KT-1000/2000 (*White et al., 1991*; *Ishii et al., 2005*), Telos (*Murer et al., 2021*; *Moser et al., 2022*; *Jung et al., 2006*), Rotameter (*Moewis et al., 2016*; *Lorbach et al., 2012*), and Rolimeter (*Schuster et al., 2011*) devices, all generally combined with stress radiography, have been presented. Studies using these techniques have demonstrated that passive knee laxity plays a key role for functional outcomes and patient satisfaction after TKA (*Oh et al., 2015*; *Jones et al., 2006*). Nevertheless, it remains unclear whether self-reported instability after TKA is reflected in implant kinematics in vivo during functional activities of daily living. Specifically, both downhill walking and stair descent are considered as challenging tasks for subjects with knee pathologies (*Stacoff et al., 2005*; *Simon et al., 2018*), and could therefore present relevant activities for provoking feelings of instability.

As the primary actuators of the locomotion system, muscles play a critical role in guiding motor functionality as well as supporting stability of the knee joint. Muscular deficits are often observed after TKA and closely correlate with physical impairment and limitations in daily activities (*Berman et al., 1991*; *Walsh et al., 1998*). Well-coordinated muscular activity is thought to be necessary for normal gait patterns (*Berman et al., 1991*). Here, adaptation of muscle recruitment patterns in the early postoperative phase, where increased knee extensor (quadriceps) and flexor (hamstrings) co-activation has generally been observed, is thought to enhance knee stability (*Lundberg et al., 2016*; *Benedetti et al., 2003*). However, whether such muscular adaptation is temporary and can be reversed in the long term after TKA remains controversially discussed (*Benedetti et al., 2003*; *Thomas et al., 2014*). Moreover, it remains unknown whether such adaptive strategies persist in unstable knees, or whether different lower limb muscle synergy patterns develop to compensate for deficits in knee stability.

While the accurate assessment of tibiofemoral kinematics during functional activities is challenging, the recent development of dynamic video-fluoroscopy systems now provides access to implant kinematics with a high level of accuracy and without soft-tissue artefact throughout consecutive cycles of gait activities (*Taylor et al., 2017*; *Schütz et al., 2019b*; *Schütz et al., 2019a*; *List et al., 2020*). Such systems have recently allowed investigations into the impact of activity (*Schütz et al., 2019a*) and implant geometry (*Schütz et al., 2019b*; *List et al., 2020*) on tibiofemoral kinematics during functional activities such as level walking, downhill walking, and stair descent, and can be combined with electromyography (EMG) to assess muscle activation synergies and its role on modulating joint kinematics (*Benedetti et al., 2003*; *Taylor et al., 2017*; *Ardestani et al., 2017*). As such, the application of these approaches to subjects with stable and unstable knees after TKA could establish whether

compensation mechanisms continue to occur, but also elucidate the role of neuromuscular activation and coordination on kinematic adaptations in the joint.

To understand tibiofemoral kinematics in unstable TKA knees and its interaction with muscle activation strategies, the objective of this study was to investigate in vivo tibiofemoral implant kinematics and muscle synergy patterns in subjects with stable and self-reported unstable knees after TKA during functional activities of daily living using dynamic video-fluoroscopy and EMG. Specifically, we aimed to establish whether unstable knees exhibit higher levels of relative tibiofemoral motion and distinct muscle synergy patterns/strategies than their stable counterparts.

## Methods

### Study cohorts

In total, 17 subjects (age ≥45 years, pain VAS ≤3) with 18 replaced knees implanted with Persona cruciate retaining (CR) TKA components and an ultra-congruent (UC) inlay (Zimmer Biomet, Warsaw, IN, USA) were recruited at least 1 year postoperatively. Both cruciate ligaments were sacrificed in all TKA knees. Of the 18, 8 TKA knees (3M:5F, 68.9 ± 8.3 years, 31.9 ± 20.4 months postoperatively, body mass index [BMI] 26.1 ± 3.2 kg/m$^2$) were recruited from subjects reporting episodes of buckling, shifting, or giving away of the operated knee during daily activities in the 3 months prior to recruitment, with or without clinical signs of instability (unstable cohort). Ten TKA knees were recruited into the stable group (7M:3F, 62.6 ± 6.8 years, 33.9 ± 8.5 months postoperatively, BMI 29.4 ± 4.8 kg/m$^2$) from subjects with no sensation of instability in the operated knee in the same 3-month period prior to recruitment. Subjects with significant problems of the lower extremities other than knee instability were excluded from participation in this study, as well as those presenting low back pain, neurological problems, patellofemoral symptoms, aseptic loosening, collateral ligament reconstruction, or an inability to perform the motion tasks, understand and/or sign the informed consent form.

The project was approved by the Zürich cantonal ethics committee (BASEC no. 2019-01242) and all subjects provided their written informed consent prior to participation.

### Clinical assessment

For each subject, clinical assessment of postoperative outcome was performed at the Schulthess Clinic Zürich, including manual passive laxity tests (anterior/posterior drawer and Lachmann, varus/valgus stress, and sagittal passive range of motion [RoM]), as well as PROMs (OKS, COMI-Knee, EQ-5D-5L, and UCLA activity score).

### Experimental procedure

Level gait (straight ahead on a level floor), downhill walking (10° inclined slope), and stair descent (three 18 cm steps) were radiographically imaged at 30 Hz using the single plane ETH Moving Fluoroscope (*List et al., 2017*), which allowed in vivo tibiofemoral kinematics to be captured throughout complete cycles of each activity. Measurement protocols for the three activities have been described previously (*Schütz et al., 2019a*; *List et al., 2020*), but are briefly described here: Prior to each motion task, trials without fluoroscopic imaging were performed until the subject felt comfortable walking with the Moving Fluoroscope. For all activities, three to five valid cycles (heel-strike to heel-strike for gait activities) were measured. Heel-strike detection was performed using eight force plates (Kistler AG, Winterthur, Switzerland, 2000 Hz, force threshold: 25 N) or heel marker trajectories (Vicon MX system; Oxford Metrics Group, Oxford, UK; 200 Hz) where no force plates were available. Muscular activations were recorded using a wireless 16-channel surface EMG system (Trigno, Delsys, USA) throughout all activities. EMG electrodes were placed on eight muscles of the TKA limb: rectus femoris, vastus medialis, vastus lateralis, semitendinosus, biceps femoris, tibialis anterior, gastrocnemius medialis, and gastrocnemius lateralis. All measurement systems were temporally synchronized. For confirmation of previous clinical assessments, subjects were asked to report any feelings of instability on a four-point scale after each activity.

### Data processing

Fluoroscopic images were distortion corrected (*Foresti, 2009*) before 2D → 3D registration was performed to determine the 3D poses of the implant components for each frame of the activity

cycles using an in-house registration software (*Burckhardt et al., 2005*) (registration errors: <1° for all rotations, <1 mm for in-plane, and <3 mm for out of plane translations; *List et al., 2017*; *Foresti, 2009*). To describe relative tibiofemoral rotations, the joint coordinate system approach reported by *Grood and Suntay, 1983* was used based on the local coordinate systems of the respective implant components. Tibiofemoral condylar A-P translations were defined based on the movement of the weighted mean location of the 10 nearest points on each condyle relative to a plane on the tibial plateau (*Schütz et al., 2019a*; *List et al., 2020*).

Kinematic parameters including tibiofemoral A-P translations, flexion/extension, ab/adduction, and internal/external rotation angles during all gait activities, were interpolated to 101 data points per cycle. Condylar A-P translations were presented relative to the corresponding medial condyle position at the initial heel-strike of each trial. For each task, the RoM of all kinematic parameters was determined and separated into stance and swing phases. The intercondylar A-P RoM (lateral RoM – medial RoM) was then used to evaluate the transverse plane pivot pattern.

EMG data were processed in R (version 4.2.0, R Foundation for Statistical Computing, Vienna, Austria) using package 'musclesyneRgies v1.2.5' (*Santuz, 2022*). All EMG signals were filtered (high-pass, cut-off frequency 50 Hz, fourth order; full wave rectified; low-pass, cut-off frequency 20 Hz, fourth order), amplitude normalized to the maximum value in each subject activity (*Santuz et al., 2017*), and time normalized to 200 data points, assigning 100 points to the stance and 100 to the swing phases (*Santuz et al., 2018b*). This time-normalization approach was chosen to ensure that the results could be interpreted independently of the absolute duration of gait events. Muscle synergy weights (time-independent coefficients), as well as the corresponding activation patterns (time-dependent coefficients), were extracted using a non-negative matrix factorization algorithm and subsequently functionally classified using *k*-mean clustering to evaluate the consistency of muscle synergies across each group (stable vs unstable) and activity (total of six classifications). Here, the number of classified muscle synergies in each group was imposed based on the average number of muscle synergies extracted per activity (4 for level gait and downhill walking, 3 for stair descent). The full width at half maximum (FWHM) and centre of activity (CoA) of each classified synergy activation pattern were further evaluated and compared between the two groups. The centre of activity, employed to estimate the timing of main activation, was calculated as the angle of the vector in polar coordinates

**Table 1.** Clinical assessment data of the stable and unstable groups shown as mean ± standard deviation of each parameter.

BMI: body mass index; PTS: posterior tibial slope; RoM: range of motion; OKS: Oxford Knee Score; COMI-Knee: Core Outcome Measures Index-Knee; EQ-VAS: EQ-Visual Analogue Scales. Bold values indicate a significant difference.

| Baseline data | Stable (*N* = 10) | Unstable (*N* = 8) | p |
|---|---|---|---|
| Sex ratio | 7M:3F | 3M:5F | 0.168 |
| Age [years] | 62.6 ± 6.8 | 68.9 ± 8.3 | 0.096 |
| BMI [kg/m²] | 29.4 ± 4.8 | 26.1 ± 3.2 | 0.113 |
| Time post-op [months] | 33.9 ± 8.5 | 31.9 ± 20.4 | 0.779 |
| PTS [°] | 82.0 ± 2.4 | 82.3 ± 3.0 | 0.863 |
| Inlay thickness [mm] | 11.3 ± 1.0 | 11.8 ± 1.7 | 0.511 |
| Knee flexion RoM [°] | 125.0 ± 7.8 | 126.3 ± 6.4 | 0.720 |
| Hyperextension | 1/10 | 8/8 | **<0.01** |
| Drawer tests | 1/10 | 5/8 | **0.019** |
| Varus/valgus stress tests | 1/10 | 5/8 | **0.019** |
| UCLA activity score | 8.3 ± 1.3 | 7.9 ± 1.1 | 0.465 |
| OKS | 46.0 ± 2.0 | 42.9 ± 3.8 | **0.040** |
| COMI-Knee | 0.2 ± 0.6 | 0.9 ± 0.8 | **0.044** |
| EQ-VAS | 84.5 ± 13.4 | 81.3 ± 17.1 | 0.660 |

that points to the centre of mass of the circular distribution defined between 0° and 360° or, in other words, between one touchdown and the next (*Cappellini et al., 2016*).

## Statistics

Statistical analysis was performed using MATLAB (R2022a, MathWorks, Natick, MA, USA) and R (version 4.2.0). Two sample *t*-tests were conducted to compare differences in kinematic parameters using Bonferroni correction to account for possible interdependencies (tibiofemoral A-P translation RoMs, rotation RoMs), as well as in FWHM and CoA of corresponding classified activation patterns between groups. Non-parametric statistics using the Chi-squared test were performed to assess differences between the two groups in sex ratio, as well as in passive hyperextension, drawer tests, and varus/valgus stress tests. One-dimensional statistical parametric mapping (*Pataky et al., 2016*) was used to test the effects of knee instability on all kinematic parameters over the time series of a gait cycle. One-way analyses of variance (ANOVAs) were used to examine the effects of knee instability on the muscle synergy weights that demonstrate significant differences in FWHM/CoA of corresponding activation patterns between the groups. For any parameters that exhibit statistical significance between the two groups, corresponding Cohen's *d* effect size (ES) was determined. Statistical significance was set to $p < 0.05$.

## Results

### Clinical assessment

Passive clinical examination revealed that one stable knee and all unstable knees showed signs of mild hyperextension (*Table 1*). One stable knee and five out of eight unstable knees exhibited increased passive sagittal and coronal laxity. No other differences in the clinical examination were observed between the two groups. Inferior OKS and COMI-Knee scores were observed in the unstable knees, even though comparable UCLA activity and EQ-VAS scores were observed between the two groups.

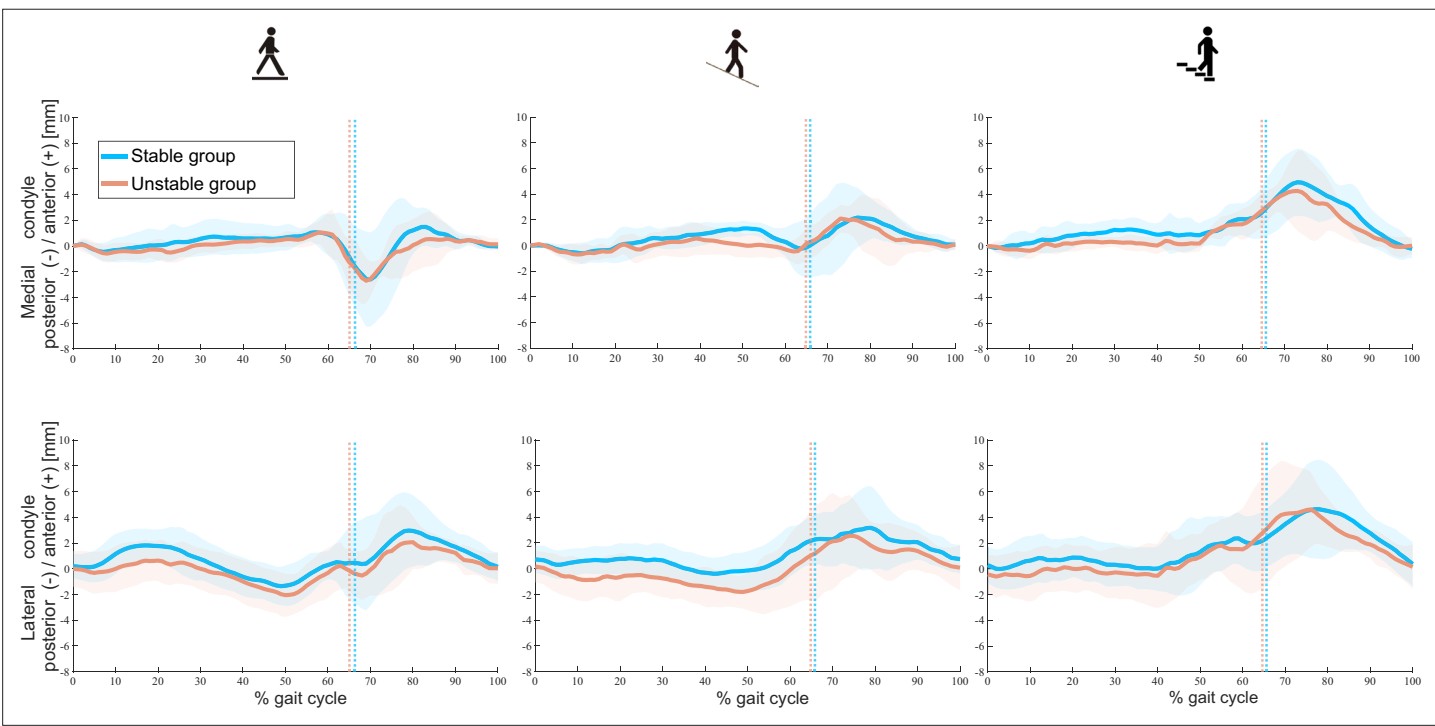

**Figure 1.** Tibiofemoral A-P translations in stable and unstable total knee arthroplasty (TKA) knees during level walking (left), downhill walking (middle), and stair descent (right). Means (solid lines) and standard deviation (shaded areas) of A-P translations in both groups are presented. Dotted colour lines indicate the mean toe-offs for the stable and unstable groups.

**Table 2.** Mean ± standard deviation of the anterior-posterior(A-P) tibiofemoral positions for the medial and lateral condyles, flexion/extension (Flex/ex), adduction/abduction (Ab/ad), and internal/external (Int/ext) rotation angles in stable and unstable groups during all activities.

| | | Stance phase | | | Swing phase | | |
|---|---|---|---|---|---|---|---|
| **A-P translation RoM [mm]** | | Medial | Lateral | Diff | Medial | Lateral | Diff |
| | Stable | 5.4 ± 1.4 | 5.1 ± 1.0 | −0.3 ± 2.2 | 7.0 ± 2.5 | 5.7 ± 1.9 | −1.1 ± 1.5 |
| Level walking | Unstable | 4.5 ± 0.9 | 4.5 ± 0.8 | −0.3 ± 1.8 | 6.2 ± 1.6 | 5.5 ± 1.5 | −0.6 ± 2.1 |
| | Stable | 4.2 ± 0.9 | 4.0 ± 0.7 | −0.2 ± 1.0 | 5.7 ± 1.5 | 4.7 ± 1.2 | −1.0 ± 1.6 |
| Downhill walking | Unstable | 3.4 ± 1.0 | 4.5 ± 0.7 | 1.1 ± 1.3 | 5.5 ± 1.5 | 5.4 ± 2.0 | −0.1 ± 1.6 |
| | Stable | 5.1 ± 1.6 | 5.5 ± 1.9 | 0.4 ± 0.8 | 7.7 ± 1.7 | 6.0 ± 2.1 | −1.7 ± 2.0 |
| Stair descent | Unstable | 4.6 ± 1.2 | 5.5 ± 1.7 | 0.9 ± 0.8 | 6.9 ± 1.7 | 7.4 ± 2.9 | 0.5 ± 2.4 |

## A-P translations

Video-fluoroscopic analysis of the functional kinematics revealed no significant differences were found in the relative A-P positions of the medial and lateral condyles at heel-strike between subjects of the stable and unstable groups (*Supplementary file 1*). Moreover, comparable mean A-P translations, A-P RoMs and pivot patterns were found between groups for the medial and lateral condyles throughout the stance and swing phases of all activities (*Figure 1* and *Table 2*). For level and downhill walking, variability was generally higher in unstable TKA knees than in their stable counterparts. Interestingly, however, unstable knees exhibited less variability in medial condylar A-P translation during late-stance and early-swing phases (≈60–75% gait cycle) compared to stable knees.

## Rotations

No significant differences in knee rotations were observed between the two groups during either the stance or swing phase for all tasks (*Figure 2*). Only the range of ab/adduction during the stance phase of downhill walking exhibited a significant difference between the cohorts (unstable 2.8 ± 0.3°, stable 2.2 ± 0.3°, $p < 0.001$, *Table 3*).

## Instability during activity

Within the unstable group, three subjects specifically reported the feeling of joint instability while undertaking the measured activities. Interestingly, all three subjects reported instability during the more challenging activities of downhill walking and stair descent, where individual kinematic outlying characteristics were observable (*Figure 3*).

## Muscle synergy analysis

In general, a comparable number of muscle synergies were extracted between stable and unstable knees during level walking (stable 3.7 ± 0.5 vs unstable 3.5 ± 0.5), downhill walking (3.6 ± 0.5 vs 3.6 ± 0.5), and stair descent (3.4 ± 0.5 vs 3.2 ± 0.9). By rounding these number of synergies, the number of classified fundamental synergies was set to 4 for level walking and downhill walking, and 3 for stair descent (*Figure 4*). The ratio of classifiable muscle synergies to total muscle synergies was slightly increased from level gait to downhill walking in the stable group (89% vs 94%) but dropped from 95% to 80% in the unstable group. The ratios were lower (stable 78% vs unstable 77%) in both groups during stair descent. During level and downhill walking, each classified synergy was dominated by the knee extensor (quadriceps), plantarflexor (gastrocnemii), dorsiflexor (tibialis anterior), or knee flexor (hamstrings) muscle groups, individually. These four distinct synergies were observable among the stable knees in a comparable manner between level walking (90%, 100%, 60%, and 90% of subjects exhibited this pattern) and downhill walking (100%, 90%, 60%, and 90%). However, these synergies

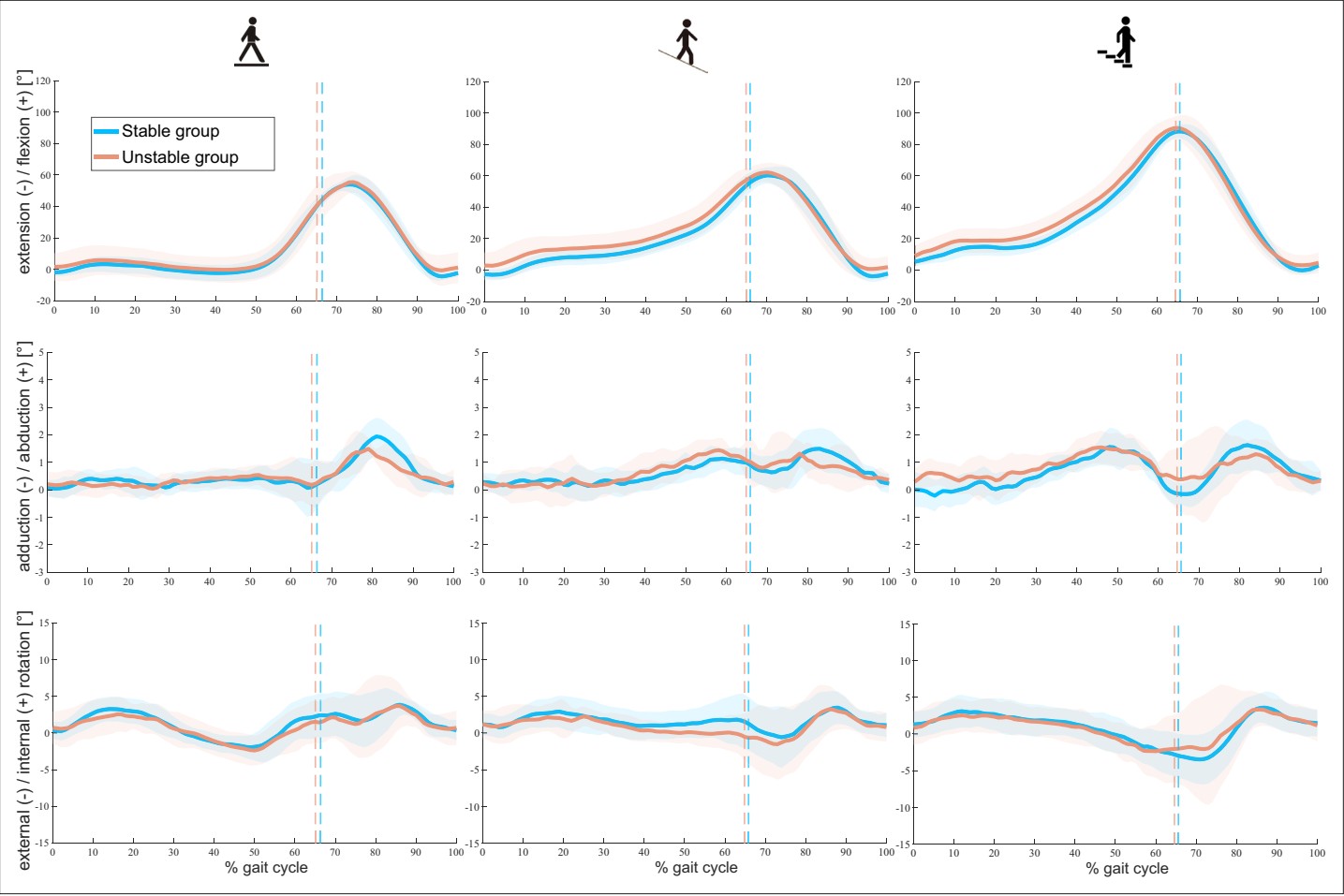

**Figure 2.** Tibiofemoral rotations throughout a gait cycle in stable and unstable total knee arthroplasty (TKA) knees during level walking (left), downhill walking (middle), and stair descent (right). Means (solid lines) and standard deviations (shaded areas) of both groups are presented. Dotted lines indicate the mean toe-offs for each group.

were less frequently observed in unstable knees during downhill walking (86%, 71%, 86%, and 43%) compared to level walking (100%, 100%, 50%, and 83%), as well as to the stable knees during downhill walking. During stair descent, three synergies, dominated by the knee extensors, dorsi- and knee flexors, or plantarflexors accordingly, could be identified in both the stable and unstable cohorts, but

**Table 3.** Mean ± standard deviation of knee range of flexion/extension (flex/ex), abduction/adduction (ab/ad), and internal/external (int/ext) rotations for the stance and swing phases of level walking, downhill walking, and stair descent.
A significant difference was observed only between stable and unstable groups in abduction/adduction during downhill walking ([a]).

| Rotation RoM [°] | | Stance phase | | | Swing phase | | |
|---|---|---|---|---|---|---|---|
| | | flex/ex | int/ext | ab/ad | flex/ex | int/ext | ab/ad |
| Level walking | Stable | 47.2 ± 4.9 | 7.7 ± 1.1 | 2.1 ± 0.2 | 61.8 ± 4.4 | 6.9 ± 1.8 | 2.8 ± 0.5 |
| | Unstable | 43.5 ± 6.8 | 7.0 ± 1.8 | 2.1 ± 0.3 | 59.6 ± 8.0 | 7.8 ± 2.5 | 2.7 ± 0.7 |
| Downhill walking | Stable | 57.9 ± 3.8 | 5.5 ± 0.9 | 2.2 ± 0.3[a] | 68.6 ± 4.6 | 7.0 ± 1.3 | 2.5 ± 0.4 |
| | Unstable | 54.9 ± 4.3 | 6.1 ± 1.2 | 2.8 ± 0.3[a] | 67.4 ± 6.7 | 8.2 ± 1.8 | 2.8 ± 0.7 |
| Stair descent | Stable | 84.4 ± 3.4 | 8.7 ± 1.2 | 3.5 ± 0.5 | 91.2 ± 4.5 | 9.4 ± 1.9 | 2.8 ± 0.6 |
| | Unstable | 84.5 ± 11.7 | 8.4 ± 1.0 | 3.3 ± 0.4 | 92.8 ± 5.8 | 10.3 ± 2.6 | 3.1 ± 0.9 |

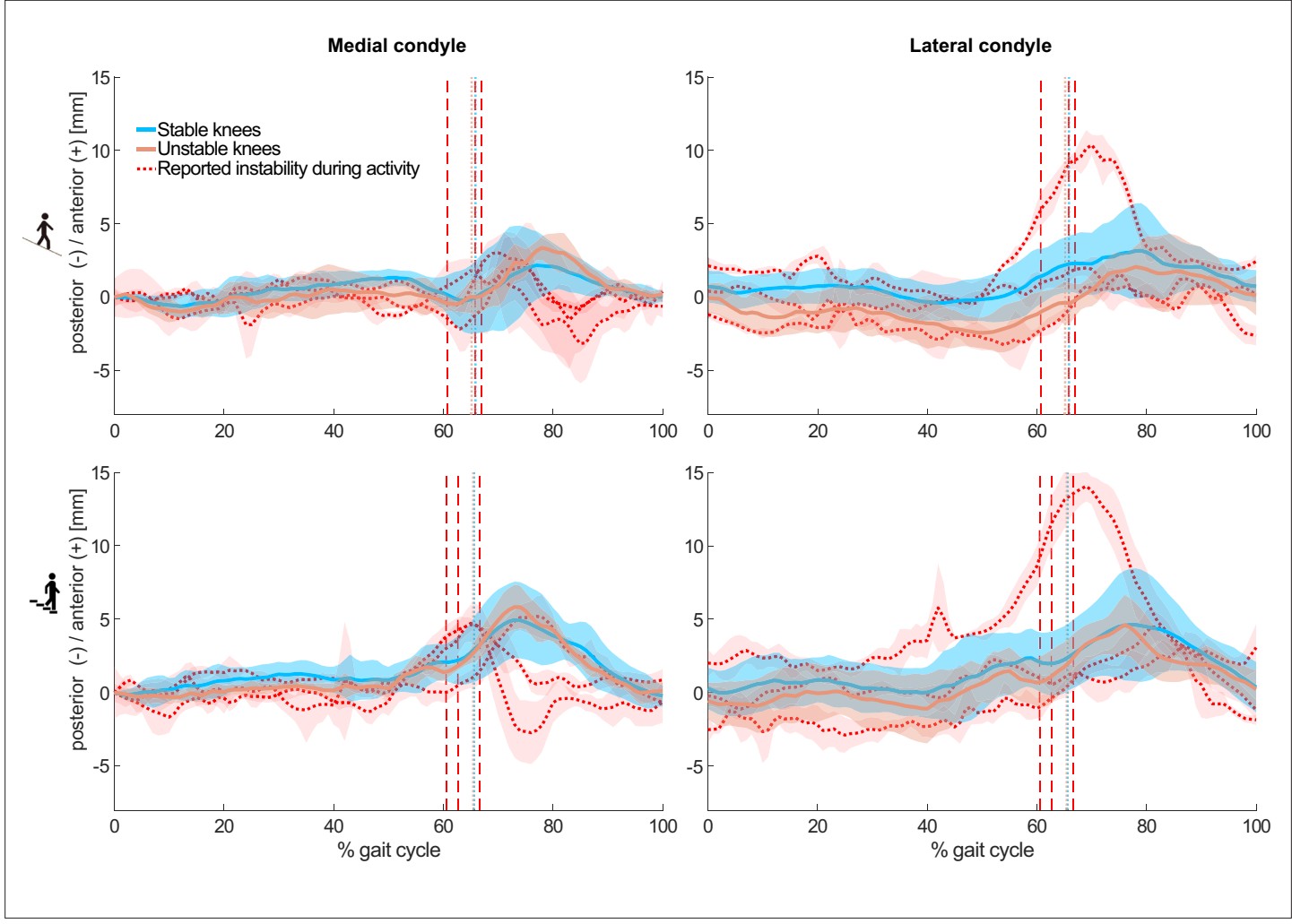

**Figure 3.** Mean and standard deviations (shaded) across subjects of the tibiofemoral A-P translations in the stable (8/8, blue), and unstable (5/8, orange) groups without reported instability during the measured activities. In addition, mean and standard deviations (shaded) across trials of three individuals from the unstable group who reported instability during the measured activities are shown in red. Dashed colour lines indicate the mean toe-offs for the stable and unstable groups, as well as for each unstable total knee arthroplasty (TKA) knee with specifically reporting instability.

were more prevalently found in the stable knees (100%, 90%, and 100%) than their unstable counterparts (100%, 75%, and 75%).

Muscle synergies exhibited phase-dependent activation patterns. Knee extensor muscles were predominantly activated during early stance (level walking) or throughout stance (downhill walking and stair descent) phases, while the plantarflexors contributed to the late stance (level walking), throughout stance (downhill walking), and mid-late swing (stair descent) phases (*Figure 4*). High contributions of the knee flexor muscles were identified during late swing/early stance (all activities), as well as at toe-off (downhill walking). As expected, the dorsiflexors exhibit a typical pattern of activation, showing strong contributions at early stance and early swing during level walking, while presenting a more sustained activation during downhill walking and stair descent.

The FWHM of the classified synergy activation patterns of the dorsi- and knee flexors were significantly higher in the unstable than the stable knees (ES = 1.49, p = 0.01) during stair descent (*Table 4*). Comparable results were found in the FWHM of the knee extensor and plantarflexor activations between the two groups during all activities. No differences were observed in the CoA of any synergies except the activation pattern dominated by dorsiflexors during level walking (ES = 2.17, p < 0.01) (*Table 5*), but the corresponding classified muscle synergy was only found in three out of the eight unstable knees. The one-way ANOVA showed similar muscle weights between the groups during level walking and stair descent (*Supplementary file 2*).

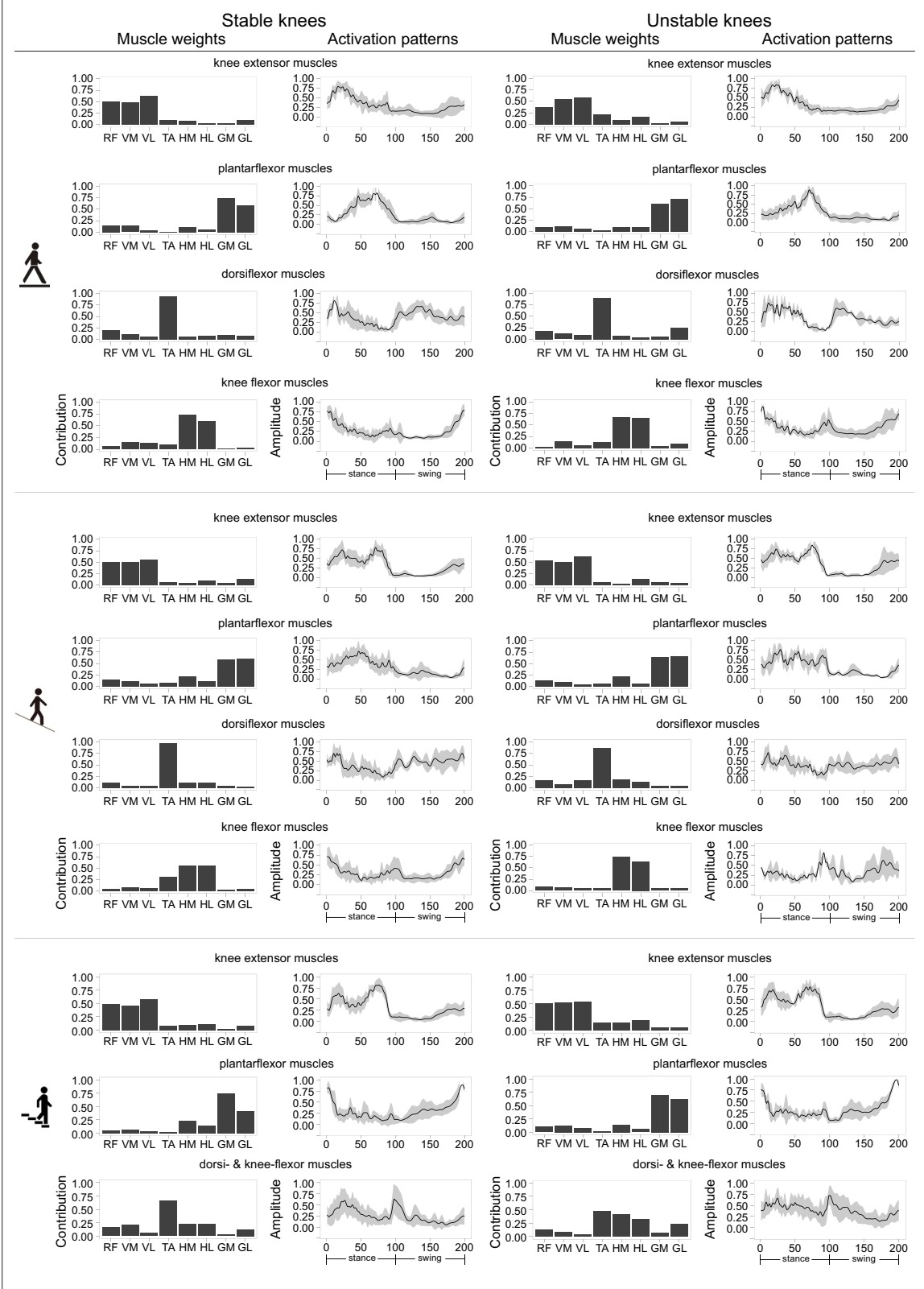

**Figure 4.** Classified muscle synergies in both stable and unstable total knee arthroplasty (TKA) knees during level walking, downhill walking, and stair descent. Muscle synergy weights, as well as means (solid lines) and standard deviations (shaded areas) of the corresponding time/amplitude-normalized activation patterns are presented for each activity. RF: rectus femoris, VM: vastus medialis, VL: vastus lateralis, TA: tibialis anterior, HM: medial hamstrings, HL: lateral hamstrings, GM: gastrocnemius medialis, GL: gastrocnemius lateralis.

**Table 4.** Mean ± standard deviation of full width at half maximum (FWHM) of the synergistic activation patterns corresponding to knee extensor, plantarflexor, dorsiflexor, and knee flexor muscle groups during level walking, downhill walking, and stair descent. Significant differences were observed between stable and unstable knees during stair descent in dorsi- ([a]) and knee flexor muscles ([b]), with an effect size of 1.49. *The classified synergy corresponding to dorsi- and knee flexor muscles was only observed in a small number of unstable knees (3/8).

| FWHM | Knee extensors | | | Plantarflexors | | | Dorsiflexors | | | Knee flexors | | |
|---|---|---|---|---|---|---|---|---|---|---|---|---|
| | Stable | Unstable | p | Stable | Unstable | p | Stable | Unstable | p | Stable | Unstable | p |
| Level walking | 29.3 ± 8.1 | 28.6 ± 13.1 | 0.55 | 24.3 ± 7.9 | 25.5 ± 6.7 | 0.38 | 37.4 ± 13.2 | 27.7 ± 8.3 | 0.11* | 21.7 ± 6.2 | 30.4 ± 9.5 | 0.06 |
| Downhill walking | 31.2 ± 9.4 | 34.1 ± 6.6 | 0.26 | 20.2 ± 7.3 | 22.6 ± 8.4 | 0.29 | 35.2 ± 15.9 | 31.1 ± 13.2 | 0.32 | 20.2 ± 5.7 | 21.6 ± 5.5 | 0.37* |
| Stair descent | 31.8 ± 12.2 | 36.0 ± 7.3 | 0.20 | 26.0 ± 9.4 | 33.5 ± 15.1 | 0.12 | 20.1 ± 8.1 | 35.0 ± 12.5 | 0.01[a] | 20.1 ± 8.1 | 35.0 ± 12.5 | 0.01[b] |

## Discussion

Knee instability is one of the most common reasons for unsatisfactory outcomes after TKA, accounting for up to 26% of revisions (*Parratte and Pagnano, 2008*; *Wilson et al., 2017*). However, the relationships between self-reported assessment of instability and knee functionality in daily living remain unclear. Using moving video-fluoroscopy and EMG, we compared kinematic parameters of knee joint stability, as well as muscle synergy patterns, between subjects who had expressed concern at the stability of their knee, and those who were happy with its functional performance. Our results indicate that kinematic parameters, including A-P translations and knee rotations, as well as their RoMs during functional activities, were generally comparable between stable and unstable TKA knees. However, increased heterogeneity in muscle synergy patterns between subjects and across activities, especially during challenging tasks such as stair descent, as well as prolonged activation of the classified knee flexor synergy, were observed in the unstable group. These differences between groups plausibly reveal long-lasting muscular adaptation strategies that develop as a compensation mechanism for feelings of joint instability and serve to prevent possible unstable events. Despite these modifications in muscular strategies, however, specific cases of acute instability were still reported by some subjects during the measurements, and analysis of their data revealed clear deviations in kinematic patterns from those of both the stable and unstable groups. These observations suggest that accurate movement analysis might be highly sensitive for detecting acute instability events, but might be less robust in identifying general joint instability.

The range of A-P translation in both groups studied here was comparable to previous studies during functional activities in good outcome TKAs with similar UC inlay geometries (*Schütz et al., 2019b*; *Roberti di Sarsina et al., 2022*; *Cardinale et al., 2020*). The so-called 'paradoxical anterior translation' pattern that is commonly observed in tibiofemoral kinematics after TKA (*Dennis et al., 2003*) was also observed in both groups during level walking and became even more evident during downhill walking and stair descent. Interestingly, while the stable and unstable groups showed comparable A-P translation RoMs on both condyles (*Table 2*), the assessment of subject-specific tibiofemoral

**Table 5.** Mean ± standard deviation of centre of activity (CoA) of the synergistic activation patterns corresponding to knee extensor, plantarflexor, dorsiflexor, and knee flexor muscle groups during level walking, downhill walking, and stair descent.
A significant difference was observed between stable and unstable knees during stair descent in dorsiflexor muscles ([a]), with an effect size of 2.17. *The classified synergy corresponding to dorsi- and knee flexor muscles was only observed in a small number of unstable knees (3/8).

| CoA | Knee extensors | | | Plantarflexors | | | Dorsiflexors | | | Knee flexors | | |
|---|---|---|---|---|---|---|---|---|---|---|---|---|
| | Stable | Unstable | p | Stable | Unstable | p | Stable | Unstable | p | Stable | Unstable | p |
| Level walking | 31.8 ± 11.7 | 34.6 ± 23.2 | 0.39 | 60.5 ± 6.4 | 57.1 ± 8.5 | 0.81 | 141.4 ± 37.3 | 68 ± 22.7 | 0.004[a]* | 34.7 ± 22.9 | 69.8 ± 78.8 | 0.19 |
| Downhill walking | 35.6 ± 6.7 | 32.6 ± 8.4 | 0.77 | 45.8 ± 10.4 | 46.4 ± 9.3 | 0.46 | 119.8 ± 46.7 | 98.3 ± 51.8 | 0.23 | 86.5 ± 39.8 | 108.1 ± 30.1 | 0.19* |
| Stair descent | 41.2 ± 8.0 | 41.4 ± 9.4 | 0.49 | 157.3 ± 27.3 | 172.4 ± 29.6 | 0.16 | 61.9 ± 24.9 | 68 ± 26.2 | 0.33 | 61.9 ± 24.9 | 68 ± 26.2 | 0.33 |

kinematics was capable of identifying acute instability events. In vivo kinematics of three unstable knees who reported instability during downhill walking and stair descent showed distinct A-P motion patterns, with an early, rapid condylar posterior translation followed by an additional anterior translation in the early/mid-swing phase (*Figure 3*). Such kinematic patterns stand out from those of both the stable cohort and even the unstable knees that did not report instability during the measured activities. Although our findings suggest that the accurate assessment of tibiofemoral kinematics has the potential to identify acute instability events, we have not been able to detect a common distinct kinematic pattern throughout the cohort of unstable knees.

Extensive efforts have been made to understand kinematic adaptations in knee pathologies including osteoarthritis (OA) and posterior cruciate ligament deficiency (PCLD). However, no significant differences have yet been found between PCLD and healthy knees in abduction/adduction or in internal/external rotation (*Fontboté et al., 2005*), nor in knee flexion/extension between OA knees and healthy controls (*Na et al., 2018*) during level gait. During more challenging activities such as stair descent, the differences in knee flexion angle and extensor torque between PCLD and healthy knees have been found to be statistically significant but presented no outstanding clinical consequences (*Hooper et al., 2002*). Interestingly, the structural deficits presented in such cases did not seem to result in clear kinematic deviations. Similarly in our study of knee joint instability, it seems that tibiofemoral movement patterns become well-compensated/controlled, plausibly through altered muscular control mechanisms.

It was previously found that unsatisfactory TKA knees present fewer muscle synergies for locomotion compared to satisfactory TKA knees and healthy controls (*Ardestani et al., 2017*). However, our results suggest that more heterogenous synergy patterns were developed in the unstable compared to the stable knees in the long term, albeit with a comparable number of synergies extracted from both groups in all activities. A higher number of non-classifiable, subject-specific muscle synergies was identified in the unstable compared to the stable group during level walking, and this number further increased during the more demanding activities of downhill walking and stair descent. This finding suggests that more challenging locomotor activities have a greater potential for provoking knee instability, which is plausibly linked to the requirement for a specific or individual response to each unstable situation or musculoskeletal deficit. Higher FWHM in the knee-/dorsiflexor-controlled activation patterns were observed in the unstable knees compared to their stable counterparts during stair descent, which further indicates that the feeling of instability can permanently trigger prolonged activation of flexor muscle groups. This finding is in line with previous studies reporting the widening of muscle activation patterns in pathologies (*Cappellini et al., 2016*; *Martino et al., 2015*; *Janshen et al., 2020*), early developmental stage (*Cappellini et al., 2016*; *Dominici et al., 2011*), ageing (*Santuz et al., 2020*; *Dewolf et al., 2021*), and presence of external mechanical perturbations (*Santuz et al., 2018a*). A possible explanation for this peculiar behaviour can be found in mouse studies, where similar temporal modifications of muscle activity have been associated with a lack of sensory feedback from proprioceptors (*Akay et al., 2014*; *Mayer and Akay, 2018*; *Santuz et al., 2019*). Interestingly, our results showed that knee extensor–flexor co-activation was present throughout the stance phase of the gait cycle in both TKA groups in all activities. However, this co-activation pattern was generally not reported in healthy stable knees (*Santuz et al., 2018a*; *Cappellini et al., 2006*; *Lacquaniti et al., 2012*). This finding shows that co-activation of knee extensors and flexors appears after TKA and persists in the long term. Moreover, we found a major involvement in TKA knees of knee flexors in the propulsion/early-swing phase during all activities, contrary to what is typically found in healthy subjects (*Santuz et al., 2020*; *Santuz et al., 2018a*; *Cappellini et al., 2006*; *Lacquaniti et al., 2012*). These activation bursts of the knee flexors possibly serve as an additional adaptive strategy to ensure joint stability during daily activities (*Benedetti et al., 2003*; *Hirokawa et al., 1991*).

There are some limitations to this retrospective observational study. Firstly, only small sample sizes were considered, which consisted of 70% male in the stable group but >60% female in the unstable group. Since subjective differences in soft-tissue properties are known to exist between males and females, it is possible that gender imbalance could have influenced the results of the passive laxity tests presented. Even though no statistical differences were found between our study groups, a more balanced cohort should be investigated in future to confirm the findings of this study. Moreover, self-reported instability is highly subjective and activity dependent, which is visible in the high inter-subject kinematic variability, and not all patients in the unstable group reported a feeling of instability during

measurement. Our analyses have, therefore, not been able to capture all the characteristics of self-reported instability, but have rather been able to make progress in understanding the relationships between several important confounding factors, as well as the individual compensation mechanisms that have developed due to joint instability. The unstable knees in this study demonstrated highly heterogenous, unclassifiable muscle synergies with variable activation patterns, suggesting patient-specific muscular strategies. However, it remains unclear whether the observed synergy patterns are a hallmark for classifying knee stability. Lastly, only a single implant type was examined in this study. Here, we acknowledge that considerable further investigation would be required to establish whether the same or similar results are observed in subjects with other implants. Moreover, it is likely that more challenging tasks (e.g. stop-and-go activities) could expose the subjects to more frequent instability events than the daily activities investigated in our study, but such tasks are hard to standardize for reliable assessment of kinematic instability.

## Conclusion

This study quantitatively investigated the tibiofemoral kinematics and muscle synergy patterns in stable and unstable TKA knees during functional activities of daily living using moving video-fluoroscopy and EMG. Our results showed that condylar A-P translations, rotations, as well as their RoMs were comparable between stable and unstable TKA knees. However, subjects who reported instability events during measurement showed distinct, subject-specific tibiofemoral kinematic patterns that differed from those in stable knees. More importantly, the normal tibiofemoral kinematics observed in the unstable group were accompanied by a greater heterogeneity in muscle synergy patterns and prolonged activation of knee flexor muscles compared to the stable group. These differences between groups plausibly reveal long-lasting muscular adaptation strategies that develop as a compensation mechanism for feelings of joint instability and serve to prevent possible unstable events. Our findings suggest that the analysis of muscle synergies is able to identify muscular adaptation that likely results from feelings of joint instability – and is therefore indicative of underlying chronic knee instability – whereas tibiofemoral kinematics are sensitive for detecting acute instability events during functional activities.

## Acknowledgements

The authors would like to thank Michael Plüss, Samara Stulz, Dr Barbara Postolka, Julia Kündig, Alexander Mattmann, and Peter Schwilch for supporting the construction of the experimental setup, as well as the lab measurements. We are also grateful to Zimmer Biomet for providing the CAD models of the Persona implant components.

## Additional information

### Funding

| Funder | Grant reference number | Author |
|--------|------------------------|--------|
| Orthopedic hospital DongXiang | External research fellowship | Longfeng Rao |

The funders had no role in study design, data collection, and interpretation, or the decision to submit the work for publication.

### Author contributions

Longfeng Rao, Conceptualization, Data curation, Formal analysis, Funding acquisition, Validation, Investigation, Visualization, Methodology, Writing - original draft, Writing – review and editing; Nils Horn, Conceptualization, Resources, Data curation, Supervision, Investigation, Methodology, Writing – review and editing; Nadja Meister, Data curation, Formal analysis, Validation, Investigation, Writing – review and editing; Stefan Preiss, Conceptualization, Resources, Supervision, Investigation, Methodology, Project administration, Writing – review and editing; William R Taylor, Conceptualization, Resources, Supervision, Validation, Visualization, Methodology, Project administration, Writing

– review and editing; Alessandro Santuz, Software, Formal analysis, Validation, Visualization, Methodology, Writing – review and editing; Pascal Schütz, Conceptualization, Resources, Software, Supervision, Validation, Investigation, Visualization, Methodology, Writing – review and editing

### Author ORCIDs
Longfeng Rao ⓘ http://orcid.org/0000-0002-2139-4972
William R Taylor ⓘ http://orcid.org/0000-0003-4060-4098
Pascal Schütz ⓘ http://orcid.org/0000-0003-1711-7881

### Ethics
The project was approved by the Zürich cantonal ethics committee (BASEC no. 2019-01242), and all subjects provided their written informed consent prior to participation.

### Decision letter and Author response
Decision letter https://doi.org/10.7554/eLife.85136.sa1
Author response https://doi.org/10.7554/eLife.85136.sa2

## Additional files

### Supplementary files
• Supplementary file 1. *Mean ± standard deviation of kinematic parameters of interest in stable and unstable groups at the instant of heel-strike in all activities. Medial A-P: anterior–posterior tibiofemoral positions for the medial condyles; Lateral A-P: anterior–posterior tibiofemoral positions for the lateral condyles; flex/ex: flexion/extension angles; ab/add: adduction/abduction angles; int/ext: internal/external rotation angles.*

• Supplementary file 2. Post hoc pair-wise comparisons of one-way analysis of variance (ANOVA) results on hamstrings-dominant classified synergy module during stair descent. Bold values indicate the comparison of the same muscle between stable and unstable groups. RF: rectus femoris, VM: vastus medial, VL: vastus lateral, TA: tibialis anterior, HM: hamstrings medial, HL: hamstring lateral, GM: gastrocnemius medial, GL: gastrocnemius lateral.

• MDAR checklist

• Reporting standard 1. Strobe checklist.

### Data availability
Source Data files and related codes have been provided for all Figures and Tables in the supplementary and can be found here: https://doi.org/10.3929/ethz-b-000584582.

The following dataset was generated:

| Author(s) | Year | Dataset title | Dataset URL | Database and Identifier |
|---|---|---|---|---|
| Rao L | 2022 | Comparable in vivo joint kinematics between self-reported stable and unstable knees after TKA can be explained by muscular adaptation strategies | https://doi.org/10.3929/ethz-b-000584582 | ETH Library research collection, 10.3929/ethz-b-000584582 |

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
