## [Editor Report]

This paper presents a new method for evaluating joint instability at the knee after total knee replacement. The work is a valuable contribution that is based on solid evidence. The results should be of particular interest to those who study this important clinical problem.

---

## [Decision Letter]

**Decision letter after peer review:**

Thank you for submitting your article "Comparable in vivo joint kinematics between self-reported stable and unstable knees after TKA can be explained by muscular adaptation strategies: a retrospective observational study" for consideration by *eLife*. Your article has been reviewed by 2 peer reviewers, and the evaluation has been overseen by a Reviewing Editor and Mone Zaidi as the Senior Editor. The following individuals involved in the review of your submission have agreed to reveal their identity: Christopher Cardozo (Reviewer #1); Scott Banks (Reviewer #2).

Essential revisions:

Please carefully consider the comments of each of the reviewers and prepare a revised manuscript that addressed them. Please include with your revision a point-by-point response to these comments.

*Reviewer #1 (Recommendations for the authors):*

I commend the authors on a carefully prepared and thoughtful presentation of their data. My only comment is a recommendation that they review the introduction to improve clarity and brevity. There are a few sentences with complex construction using parentheses and other punctuation that are difficult to read.

*Reviewer #2 (Recommendations for the authors):*

I was excited to read this study. The authors address an important issue for patients with TKA, joint instability. The unique imaging setup available for the study provides observations that cannot easily be reproduced elsewhere.

I believe the fact that the stable group was 70% male and the unstable group >60% female is a major limitation. Surgeons commonly report a subjective difference in the soft-tissue properties between male and female patients, so it may be necessary to have more closely matched groups to have a clinically meaningful comparison.

The condylar A-P translations were referenced to the pose at the initial heel strike. This is a common method to achieve better alignment of kinematic variables to highlight the motion patterns, but it masks the ability to determine if there was an overall bias in the condylar positions relative to the tibial articulation. For example, did the unstable knees exhibit generally more posterior or anterior contact than the stable knees? The lack of geographic context will make it difficult to compare these data to other reports using similar methods.

The methods report how the transverse plane pivot location was determined, but I did not see that parameter reported in the Results nor was it discussed.

Independent sample t-tests for A-P translations and knee rotations ignore the fact that there are just two rigid bodies moving that are parameterized with 6 DoF. For example, it would be hard to argue that the difference in condylar A-P translations is independent of the measured knee internal/external rotation.

The first three sentences of the Results section have multiple internal inconsistencies. Or it is just not clear what the authors are trying to say. In either case, this section needs to be redone.

Table 1 omits the subject sex, which is an important covariate. Further, the label 'Hypertension' should be 'Hyperextension'. And couldn't the hyperextension, drawer, and laxity tests be compared statistically using the Z-test or chi-square test?

The fact that quad/ham co-activation was observed throughout the gait cycle in both groups is a very interesting finding!

The authors are to be congratulated for conceiving this combined kinematic and EMG analysis to better understand TKA biomechanics and patient perceptions of instability. The methodology and perspectives are excellent. I am looking forward to subsequent work with bigger, better-matched cohorts and with a range of TKA designs!

---

## [Author Response]

Reviewer #1 (Recommendations for the authors):I commend the authors on a carefully prepared and thoughtful presentation of their data. My only comment is a recommendation that they review the introduction to improve clarity and brevity. There are a few sentences with complex construction using parentheses and other punctuation that are difficult to read.

The authors appreciate the positive feedback from the Reviewer. The Introduction section has now been reviewed and improved where appropriate, as suggested. Specifically, long and complex sentences with parentheses and punctuations have been shortened or rewritten in separate, shorter sentences to improve clarity and brevity.

Reviewer #2 (Recommendations for the authors):I was excited to read this study. The authors address an important issue for patients with TKA, joint instability. The unique imaging setup available for the study provides observations that cannot easily be reproduced elsewhere.

The authors appreciate such a positive reception from the Reviewer.

I believe the fact that the stable group was 70% male and the unstable group >60% female is a major limitation. Surgeons commonly report a subjective difference in the soft-tissue properties between male and female patients, so it may be necessary to have more closely matched groups to have a clinically meaningful comparison.

The authors acknowledge that gender differences between the groups could have played a role on the results of the passive laxity tests presented. However, we performed a Chi-squared test on the sex ratio between the stable and unstable groups and found a corresponding *p* value of 0.168 (Table 1), which would not demonstrate a significant association between gender and knee instability. However, the Reviewer’s point is well taken, and while we are now unable to go back and re-test subjects, we have acknowledged this possible bias in the limitations section of the discussion:

“Firstly, only small sample sizes were considered, which consisted of 70% male in the stable group but >60% female in the unstable group. Since subjective differences in soft-tissue properties are known to exist between males and females, it is possible that gender imbalance could have influenced the results of the passive laxity tests presented. Even though no statistical differences were found between our study groups, a more balanced cohort should be investigated in future to confirm the findings of this study.”

The condylar A-P translations were referenced to the pose at the initial heel strike. This is a common method to achieve better alignment of kinematic variables to highlight the motion patterns, but it masks the ability to determine if there was an overall bias in the condylar positions relative to the tibial articulation. For example, did the unstable knees exhibit generally more posterior or anterior contact than the stable knees? The lack of geographic context will make it difficult to compare these data to other reports using similar methods.

This is an interesting aspect raised by the reviewer, and we agree that the initial condylar position relative to the tibial articulation at heel strike should not be neglected. As a result, we have now calculated the corresponding data, which are presented in a new Table in the supplementary material. The data suggest that the relative tibiofemoral positions at heel strike are generally similar (no significant differences) between the stable and unstable groups, which has also now been briefly noted in the Results section (A-P translation section):

“Video-fluoroscopic analysis of the functional kinematics revealed no significant differences were found in the relative A-P positions of the medial and lateral condyles at heel strike between subjects of the stable and unstable groups. Moreover, comparable mean A-P translation patterns were found between groups for the medial and lateral condyles throughout the stance and swing phases of all activities (Figure 1, Table 2).”

In light of these results, we are confident that this important aspect is not an influencing factor governing possible differences between the stable and unstable groups.

The methods report how the transverse plane pivot location was determined, but I did not see that parameter reported in the Results nor was it discussed.

The authors apologise for this misunderstanding, which was due to ambiguity in the presented definitions. The intercondylar A-P RoM was calculated as the lateral condyle RoM minus the medial condyle RoM. These values were used to assess the transverse plane pivot pattern rather than any specific location of the centre of rotation in the transverse plane. If the intercondylar A-P RoM was positive, then a medial pivot movement pattern of the femur related to the tibial was assumed, while negative values indicated a lateral pivot pattern. Values for the transverse plane pivot pattern are now reported in Table 2 (column “Diff”), together with a short Discussion section:

“Moreover, comparable mean A-P translations, A-P range of motions and pivot patterns were found between groups for the medial and lateral condyles throughout the stance and swing phases of all activities (Figure 1, Table 2).”

Independent sample t-tests for A-P translations and knee rotations ignore the fact that there are just two rigid bodies moving that are parameterized with 6 DoF. For example, it would be hard to argue that the difference in condylar A-P translations is independent of the measured knee internal/external rotation.

The authors thank the Reviewer for this suggestion in statistical analysis. Independent t-tests alone are indeed not suitable for checking the statistical differences between two groups where interdependencies exist. Therefore, Bonferroni correction has now been used in the two-sample t-tests for all kinematic parameters of interest (medial/lateral A-P translations, δ A-P translations, flexion/extension, ab/adduction, and int/extension). Corresponding corrections have therefore also been made in the manuscript:

“Two sample t-tests were conducted to compare differences in kinematic parameters using Bonferroni correction to account for possible interdependencies (tibiofemoral A-P translation RoMs, rotation RoMs), as well as in FWHM and CoA of corresponding classified activation patterns between groups.

Video-fluoroscopic analysis of the functional kinematics revealed no significant differences were found in the relative A-P positions of the medial and lateral condyles at heel strike between subjects of the stable and unstable groups (Supplementary material S2). Moreover, comparable mean A-P translations, A-P range of motions and pivot patterns were found between groups for the medial and lateral condyles throughout the stance and swing phases of all activities (Figure 1, Table 2).”

The first three sentences of the Results section have multiple internal inconsistencies. Or it is just not clear what the authors are trying to say. In either case, this section needs to be redone.

The authors appreciate this comment on improving clarity – indeed the point regarding hyperextension was inaccurate, and the first paragraph of the Results section has therefore now been rewritten – please see the changes made within the updated manuscript.

Table 1 omits the subject sex, which is an important covariate. Further, the label 'Hypertension' should be 'Hyperextension'. And couldn't the hyperextension, drawer, and laxity tests be compared statistically using the Z-test or chi-square test?

The authors thank the Reviewer for this suggestion in statistical analysis. We’ve added the following description in the statistics part of the methods section (Page 6, last paragraph):

“Non-parametric statistics using the chi-squared test were performed to assess differences between the two groups in sex ratio, as well as in passive hyperextension, drawer tests, and varus/valgus stress tests.”

In addition, the corresponding results are now presented in Table 1 of the Results section.

The fact that quad/ham co-activation was observed throughout the gait cycle in both groups is a very interesting finding!The authors are to be congratulated for conceiving this combined kinematic and EMG analysis to better understand TKA biomechanics and patient perceptions of instability. The methodology and perspectives are excellent. I am looking forward to subsequent work with bigger, better-matched cohorts and with a range of TKA designs!

The authors appreciate this positive feedback. We are excited in further applying the methodology in this manuscript to larger patient cohorts with various TKA implant designs, and it will be interesting to see whether the results from future studies show similar outcomes.